# John Dewey's Radical Temporalism

**Vincent Colapietro**

Department of Philosophy, Pennsylvania State University, State College, PA 16801, USA; vxc5@psu.edu

**Abstract:** The author presents John Dewey's mature account of temporal continuity, showing how Dewey's position can be identified as a form of radical temporalism. Even at the most elemental level (that of subatomic particles), natural existence is for such a temporalist an irreducibly temporal affair. While he focuses primarily on Dewey's "Time and Individuality" (1940), the author supplements his account by drawing upon *Experience and Nature* (1925), "Events and the Future" (1926), and to a lesser extent, other texts. In his *magnum opus*, Dewey draws a crucial distinction between temporal quality and temporal seriality. In an essay published the following year, he insists, contra C. D. Broad, on qualitative heterogeneity being an intrinsic trait of even the thinnest slice of the tem-poral continuum. The most elemental units of the temporal flux are neither "eternal and immu-table", nor qualitatively homogenous. They also exhibit a from-to-through structure, with the dimensions of time being defined by these terms (pastness as *from which*, futurity as *toward which*, and presentness as *through which*). By relating the distinction between temporal quality and temporal seriality as well as the central claims made in "Events and the Future" to "Time and Individuality", the author brings into clear focus the contours of Dewey's radical temporalism.

**Keywords:** continuity; immediacy; individuality; ineffability; intelligibility; quality (including qualitative immediacy); temporal quality versus temporal seriality

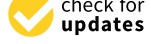



## 1. Introduction

In 1940, John Dewey published one of his most important essays. Originally given as a lecture on 21 April 1938, at New York University, "Time and Individuality" appeared in *Time and Its Mysteries* (Series II). [1] While it has hardly been ignored by scholars, [1] this essay still has not received the attention it merits. It is indeed high time to accord this late reflection a painstaking consideration. Dewey was almost eighty years old when he wrote this piece. He appears to have taken the invitation to contribute to this series of lectures as an opportunity to catch up more fully to the implications of his own thinking about time. In doing so, he took himself to be catching up to the implications of the distinctively modern concept of an *event* [7] (pp. 62–68). It should not be an embarrassment for philosophers to confess that they do not know fully or even adequately what they mean, since at the heart of their vocation is just this discovery, one that is never finally or decisively won [8] (p. 77). In any event, what might be called Dewey's radical temporalism was designed to replace traditional substantialism [3,6]. At the root of its being, every natural existent is a temporal affair: it is an instance of becoming [9] (Chapter 11).

My procedure here is primarily to explicate and contextualize Dewey's essay, but also to suggest some of the ways in which his position accords with the cutting edge of later twentieth century and early twenty-first century physicists [10] and other theorists devoted to address "the mysteries of time" [1] Cf. [11]). As C.S. Peirce suggested, in effect speaking for all pragmatists, the best any philosopher can ever do is to put forth a conjecture, "not devoid of all likelihood", moreover, "in the general line of growth of scientific ideas" [12] (p. 1, para. 7). Optimally, such a such conjecture would assist in preparing the way for generating ideas "capable of being verified or refuted by future observers". Philosophy is accordingly no less conjectural than science; it, too, is ultimately evaluated in terms of its capacity to illuminate the disclosures of our experience and, beyond this, at least to

intimate possibilities of such disclosures beyond anything presently available.[2] Heuristic fecundity, not just empirical adequacy, is a desideratum of the philosopher no less than the scientist. In the essay under discussion here, Dewey *does* what Peirce advises, as he offers a conjecture about time, "not devoid all likelihood", that is arguably in line with the emerging lines of scientific discovery. His paper is at once experiential and speculative (or theoretical). It is also, in a sense to be explained, clearly programmatic.

## 2. A Programmatic Essay

At the very least, Dewey in "Time and Individuality" touched upon metaphysical and cosmological issues without ever losing sight of the human motives and stakes entangled with these issues [14] (Chapter 1). In several critical respects, he did far more than touch upon these issues. He delved into them deeply, outlining a naturalistic understanding of the "temporal quality" of natural affairs (such affairs being for him what above all deserves to be recognized as real or existent). Even so, "Time and Individuality" is for the most part a programmatic essay, for in it, Dewey sketches a program to be carried forward into the indefinite future. Significantly, he initially looks back for the sake of going forward. We cannot understand the import of his resignification—his effort to show how a traditional signifier (*event*) has actually undergone a radical alteration, without the implications of that alteration being fully appreciated—without turning to other texts. In the first instance, however, "Time and Individuality" can serve as our entrance into his temporalism.

Bringing a historical awareness to the demanding task of simply outlining such an approach, Dewey readily acknowledged the contribution of two predecessors in particular. Henri "Bergson and William James, animated by different motives and proceeding by different methods, ... installed change at the very heart of things" [1] (pp. 100–101). Bergson took his stand on the primacy of life and consciousness, which are notoriously in a state of flux" [1] (p. 101).[3] In contrast, the "animating purpose of James was ... primarily artistic and moral" [1] (p. 101). For both theorists, a universe in which "there is no room for novelty and adventure" is one radically at odds with the world disclosed in our experience. It is, however, in connection with James that Dewey introduces the other term in his conjunctive title ("Time and Individuality"). In a "block universe" (to use James's "term of adverse criticism"), "the individual is simply a part determined by the whole of which he is a part" [1] (p. 101). This amounts to nothing less than the effacement of individuality.

In Dewey's judgment, the robust acknowledgment of irreducible individuality demands an equally decisive commitment to temporal flow (or seriality). That is, it requires radical temporalism.[4] Such temporalism encompasses plurality, indeterminacy, and of course, alterability. "Only a philosophy of pluralism, of genuine indetermination, and of change which is real and intrinsic", Dewey following James, "gives significance to individuality".

While Bergson and James were arguing for such points "in the late nineteenth century", it was not until the twentieth century that we encounter "the out-and-out assertion in systematic form that process *is* reality" [1] (p. 101). Dewey is of course referring to the contribution of A.N. Whitehead. As he reads Whitehead, however, the author of *Process and Reality* "also held that there is a fixed order which controls the ebb and flow of the universal tide" [1] (p. 101). That is, fluency is ultimately in the control of fixity, the observable flux of natural events is a site wherein the putative ingression of "eternal objects" renders time intelligible. On this occasion, Dewey barely hints at his misgivings regarding this feature of Whitehead's account of process. From other writings, however, we know of his deep disagreement with certain features of the Whiteheadean framework.

After invoking the names of Bergson, James, and Whitehead, Dewey turns in earnest to the task at hand, giving a programmatic account of time *and* individuality. "My theme ... is not historical, nor is it to argue in behalf of any one of the doctrines regarding time that have been advanced" by any of his predecessors or contemporaries. Even so, historical consciousness is especially apposite here: "The purpose of the history just roughly sketched is to indicate that the nature of time and change has now become in its own right a

philosophical problem of the first importance" [1] (p. 102). His purpose in this essay is to address "time as a problem", though not in all respects. "The aspect of the problem" in which he is principally interested "is *the connection of time with individuality*, as the latter is exemplified in living organisms and especially in human beings" [1] (p. 102; emphasis added). While he is focally concerned with individuality as it is manifested in the lives of organisms, especially humans, there are at least hints that he takes temporal quality to be bound up with the individual character of all natural existents.

Dewey invites his readers to consider "the account of any person, whether the account is a biography or an autobiography". What has become a central concern in contemporary thought (see, e.g., [15,16]) was undramatically introduced by Dewey in "Time and Individual". "The *story* begins", he notes, "with birth, a temporal incident; it extends to include the temporal existence of parents and ancestors" [1] (p. 102; emphasis added). That is, the story does *not* actually begin with the birth of the individual; it begins with events and individuals leading up to birth. The birth of the individual alone makes these antecedent events and other individuals relevant, but signals (to use C.S. Peirce's expression) "a *process of beginning*" [17] (p. 27). Beginnings are not instantaneous; rather they are, however imperceptibly, incremental or infinitesimal. Just as an adequate account of a life does not begin with birth, it "does not end with death, for it takes in the influence upon subsequent events of the words and deeds of the one whose life is told" or narrated. Any individual life "is an extensive event; or, if you prefer, it is a course of events each of which takes up into itself something of what went before and leads on to that which comes after" [1] (p. 102). That *from* which a life comes and that *toward* which it flows are integral to that *through* which the individual, as an individual, moves.[5] Neither the relevant antecedents nor the significant impacts of an individual life are necessarily obvious or even immediately legible. "The skill, the art, of the biographer is displayed in his ability to discover and portray the subtle ways, hidden often from the individual himself, in which one event grows out of those which preceded it and enters into those which follow" [1] (p. 102).

For Dewey, the conclusion to be drawn from such considerations is itself obvious: "The human individual is himself a history, a career, and for this reason his biography can be related only as a temporal event" [1] (p. 102), albeit one of an extensive character. It must take the form of a narrative in which the future tends to reconfigure the past at least as much as the past shapes the present.[6] On Dewey's view, then, individuals do not have histories; rather they *are* histories. What might not be evident, however, is one of the most important implications of this radical temporalism. "That which comes later explains", Dewey suggests, "the earlier quite as truly as the earlier explains the later" [1] (102). The subsequent as much as the antecedent has explanatory power. His example helps us grasp this point. "Take the individual Abraham Lincoln at one year, at five years, at ten years, at thirty years of age, and imagine everything later wiped out, no matter how minutely his life is recorded up to the date set" [1] (p. 102). The *significance* of what befell Lincoln at one, five, or ten years of age is to an incalculable degree a function of who this individual became at a later phase. Individuals do not move through time, as though temporality is extrinsic to their individuality. As Dewey puts it, specifically in reference to the example of Lincoln, "he did not exist in a time which externally surrounded him, but time was the heart of his existence" [1] (p. 102) as an individual. That is, time vis-à-vis individuals is not an external framework in and through which they move; it is, rather, an intrinsic quality of any unique individual (strictly speaking, "unique individual" is a pleonastic expression since individuals are, in Dewey's judgment, by definition unique).

He, however, seems to compromise his position by acknowledging what he is too good an empiricist to disregard: "Everything that can be said contrary to this conclusion is but a reminder that an individual may lose his individuality, for individuals become imprisoned in routine and fall to the level of mechanisms" [1] (p. 112). How are we to understand this observation? Is it the case that individuals can completely lose their individuality or is it rather than they can become less markedly individual, by becoming more mechanically routinized or habituated? I am strongly disposed, given the incontestable principle

of hermeneutic charity, to take Dewey's claim in the weaker sense. In some instances, it certainly may seem as though genuine time "ceases to be an integral element" in the constitution of individual lives. This is, however, never completely true. Even the most predictable person possesses the capacity to surprise others, and indeed, that individual herself. Deweyan individuality points to an inexpungable source of "unpredictable novelties" [1] (p. 112; cf. [19], pp. 110–114). In other words, genuine time never ceases to be an integral part of individual beings, however hidebound with habits—and thus predictable—these individuals become. As Dewey sees it, contra Pierre Simon Laplace, "the individual is a temporal career whose future cannot be *logically* deduced from its past" [1] (p. 107). Dewey's radical temporalism drives toward affirming ours is a world shot through with contingency, indeterminacy, unpredictability, and novelty. Strictly speaking, such a thesis cannot be proven; it can only be rendered plausible.

Recall that the focus of Dewey's concern in this essay is with "time as a problem". After making his case that an individual human being such as Abraham Lincoln is "an extensive event", Dewey raises the question of whether other beings can accurately be described in this manner. "Now an important part of the problem of time is that what is true of human individuals does not seem to be true of physical individuals" [1] (p. 103). From his perspective, human beings are themselves physical individuals but, given his emergent naturalism, they exhibit empirical traits or observable functions calling for descriptions and explanations beyond anything provided by the physical sciences. (This makes his emergent naturalism a non-reductive—even an *anti*-reductive—naturalism). In a sense, they are more than simply physical. This also appears to be true of animals other than humans. "Most persons would resent", he points out, "denial of some sort of individuality to their own dogs" [1] (p. 103).

"Now, this apparently complete unlikeness [between purely physical beings and at least certain living ones] is a part of the problem of time" [1] (pp. 103–104). Dewey identifies three ways in which theorists have addressed this problem. First, there are those who "have been content to note the difference and to make it the ground for affirming a sheer dualism between humans and other things, a ground for assigning to man a spiritual being in contrast to material things" [1] (p. 104). These theorists might be called the dualists. They recognize the uniqueness of humans, but only by sacrificing the continuity between humans and other animals. "Others, fewer in number, have sought to explain away the seeming disparity, holding that the apparent uniqueness is specious".[7] These theorists might be dubbed the reductivists. Insisting on the continuity between human and other beings allegedly entails rejecting the undeniable uniqueness of the human animal.

"Of late, there have been", Dewey adds, "a few daring souls who have held that temporal quality and historical career are the mark of everything, including atomic elements, to which individuality may be attributed" [1] (104). Individuality goes all the way down. The historical career of any natural element marks it off as an individual, in some measure unlike any other event or existent. Dewey joined these "daring souls", these speculatively audacious theorists.

## 3. Temporal Quality

His account of individuality, as presented in "Time and Individuality", as rich and suggestive as that account is, must be supplemented by what he has argued in other texts. In this regard, *Experience and Nature* (1925) and "Events and the Future" (1926) are especially salient. Specifically, the discussion in Dewey's *magnum opus* of "temporal quality" and that in the essay published a year later of the intrinsic heterogeneity of even the thinnest slice of the temporal continuum greatly deepen our understanding of the position outlined in "Time and Individuality".

There is a facet of *opacity* characteristic of individuality, a point to which he alludes in "Time and Individuality". At least experientially, there is, in Dewey's account, an *opacity* to individuals. "We are", he contends, "given to forgetting that things, with our insistence upon causation and upon the necessity of things happening as they do happen, exist

just as they qualitatively are" [1] (p. 112). For purposes of explanation and, in certain contexts, even for those of description, it is tremendously advantageous to abstract from the qualitative immediacy of actual affairs (e.g., *this* earthquake occurring at *this* time, in *this* place, *this* sunflower on *this* occasion turning toward the sun, or *this* earth worm adapting to *these* novel circumstances—see, e.g., [13,21,22]. "Given a butterfly or an earthquake *as an event*, as a change, we can at least in theory find out and state its connection with other changes" [1] (p. 112; emphasis added). The explicable eventuality of the earthquake or the butterfly (say, the violent upheaval of geological formations or the metamorphosis of the larva into an insect with wings) does not negate the sheer immediacy—the irreducible individuality—of the event or transformation. Our investment in explanations, especially ones contributing to our capacity to alter the course of events, need not prompt us to efface individuality. Indeed, "the individual butterfly or earthquake remains just the unique existence it is. We forget in explaining the occurrence that it is only the *occurrence* that is explained, not the thing itself" [1] (p. 112). The thing itself, in its qualitative immediacy and irreducible uniqueness, *is* and, moreover, just is what it immediately is. Accordingly, even "in explaining the occurrence we are compelled to fall back on other individual things that have just the unique qualities they do have". The naturalist Dewey does not hesitate to identify the opacity of individuals as a mystery, "the mystery of things being just what they are" [1] (p. 112). While a sense of this mystery might contribute, if only indirectly, to the ceaseless quest for scientific explanations, it is more powerfully evoked and cultivated by artists. "The mystery is that the world is at it is—a mystery that is the source of all joy and all sorrow, of all hope and fear" [1] (112). Nothing ever banishes this mystery. Cf. [23] (pp. 64–65). Thus, nothing ought to be allowed to destroy our sense of this mystery. It does not mark the defeat of rationality; rather it signals the limits in which our reason lives and moves, while to some extent transcending these limits.

In *Experience and Nature* (1925), Dewey suggested: "Existentially speaking, a human individual is distinctive opacity of bias and preference conjoined with plasticity and permeability of needs and likings" [13] (p. 186). "The human individual in his opacity of bias is in so far doomed to a blind solitariness" [13] (p. 186). However, "every existence in addition to its qualitative and intrinsic boundaries [to its ineliminable opacity] has affinities and active outreachings for connection and intimate union" [13] (p. 187). The self-assertion of the individual tends to be, to some degree, an instance of self-isolation, self-insistence, an abiding refusal insofar as the individual resists being absorbed without remainder into what is other than the qualitatively identifiable, thus the temporally qualifiable, individual.

In *Experience and Nature*, Dewey draws a critical distinction. He insists:

> Temporal quality is . . . not to be confused with temporal order. Quality is quality, direct, immediate and undefinable. Order is [in contrast] a matter of relation, of defining, dating, placing and describing. It is discovered in reflection, not directly had and denoted [in experience] as is temporal quality. Temporal order is a matter or science [or less disciplined forms of experimental inquiry]; temporal quality is an immediate trait of every occurrence whether in or out of consciousness. Every event is as such a passing into other things, in such a way that a later occurrence is an integral part of the *character* or *nature* of present existence. An 'affair', *Res*, is always at issue . . . Each comes from something else and each when it comes has its own initial, unpredictable, immediate qualities, and its own similar terminal qualities. The later is never just resolved into the earlie. [13] (p. 92)

That is, the later phases are not reducible to the earlier ones. If anything, the emergent and the eventual have a greater claim to ontological status than the antecedent and the ground. The rearrangement of simple, immutable elements is a posit of the theoretical imagination. The transitions, transmutations, and transformations of complex, contingent *affairs* are deliverances of human experience. The dialectical manipulation of abstract concepts is necessary but insufficient for understanding time or anything else. The experiential disclosure of possibly ubiquitous features of natural existence is needed not only

to complement this dialectic, but also to ground discourse and inquiry in the experiential matrix in which the concrete terms alone have their full meaning.

What human experience discloses about natural affairs is this. "The real existence is", as Dewey puts in in *Experience and Nature*, "the history in its entirety, the history *as just what it is*" [13] (p. 210; emphasis added). Its qualitative individuality is irreducible, not least of all to its ineliminable role in a heuristic endeavor. *To be* is not reducible to being known or to be knowable. There is more to being than cognizability, although affairs to some extent avail themselves to being more fully comprehended than what any finite encounter or series of such encounters has disclosed, to date. This is especially true if human consciousness becomes fixated on the qualitative traits of unique affairs, that is, if it does not make quantifiable relations a focal concern of experimental inquiry.[8] The objects of science are inherently general, whereas those of our experience, in its unreflective form, are qualitatively unique. Cf. [24] (p. 96; also, p 101). The objects of science are also essentially relational, while the macroscopic objects of unreflective experience are related to one another in a much more severely circumscribed manner than those of experimental inquiry. The objects of science are *not* conceivable apart from the context of a practice, though the context in question is in crucial respects wider than that of the more narrowly circumscribed context of everyday engagement with natural affairs (the context in which things are primarily *had*, rather than known or investigated[9]).

Though inherently ineffable, qualitative individuality identifies a pivot around which inquiry can turn. Cf. [13] (p. 265); also [7,22].[10] Though essentially relational and abstract, scientific intelligibility relies on qualitative discrimination and description, identification, and indeed re-identification (or resignification). Qualitative differentiation and quantifiable correlations work hand in hand to realize such intelligibility.

The life of humans might of course be unlike other affairs encountered in nature. Rather than countenance such a dualism, however, Dewey argues for the continuity between the coming to be of, say, Abraham Lincoln and that of a solar system, or even a chemical compound. The theorical imagination has been held captive by the vision of immutable elements combining according to invariant laws, at all levels. What is putatively true at the atomic level has been taken to be true at all other levels, including that of human behavior. For many, intelligibility demands a commitment to the vision of irreducibly simple elements ("atoms") being governed by effectively eternal laws. Dewey is struggling to liberate the theoretical imagination from what he takes to be an outdated vision. The *history* of physics itself points beyond any ideal of intelligibility in which simplicity, immutability, reducibility, and arguably ahistoricality are part of the very meaning of intelligibility.

## 4. From-toward-through: The Phasic Heterogeneity of Even the Thinnest Slice of the Temporal Continuum

On this occasion, I cannot do more than make the point most pertinent to "time as a problem". To do so, I must turn to "Events and the Future" as a text in which Dewey in effect supplements what he argues in *Experience and Nature* (1925) and "Time and Individuality" (1940 [1938]). This essay assists us in obtaining a deeper understanding of what he intends by "temporal quality" in contrast to "temporal order" and, indeed, of related points.

In "Events and the Future", Dewey uses C.D. Broad as a foil for his own position. Specifically, Broad might be said to allow a focus on temporal *order* (or seriality) only then to eclipse any consideration of temporal *quality*. This suggests that Dewey's embrace of temporality is more radical and unqualified than anything allowed by Broad's conceptualization of time.[11] He quotes a passage in which Broad in effect identifies "an indispensable character of anything which may be termed an event".[12] This is "qualitative variation of parts with respect to the whole which requires duration in which to display itself" [7] (p. 62). As it turns out, however, Broad "does *not* regard qualitative variation to be involved in the definition of event" [7] (p. 63). In Dewey's judgment, this makes an *event* in Broad's sense an unfortunate holdover from a prescientific epoch. Since events can be divided into slices in which there is no trace of temporality, there is on Broad's

understanding no intrinsic trait of temporal quality. To show this, Dewey quotes Broad's text: "there may well be objects which are temporally homogenous. This would mean that, however you divide up their *history*, the contents of the slices are the same as each other and as the whole in quality" [7] (p. 63; emphasis added by Dewey). Broad does not hesitate to draw from this consideration the conclusion that science regards its own *ultimate* objects as "spatio-temporally homogenous"; and that means science assumes its own ultimate objects "never began or end"; and, finally, that "the ultimate scientific objects are regarded as eternal in the sense of existing throughout all time" [7] (p. 63); [5] (p. 403).

"Eternal objects", Dewey is quick to note, "have no 'history', much less a history which can be 'divided'". The reason is plain, but he nonetheless makes it explicit: "Bare persistence is not history and it has no stages; the moment they are referred to as stages qualitative change is introduced" [7] (p. 64). Temporal quality is the intrinsic qualitative variation of anything (after the epochal *changes* in scientific thought) properly designated as an "event". "Events" in Broad's sense are, in Dewey's judgment, simply not events in the scientific sense, if the history of science (specifically, such episodes as relativity and quantum theory) are taken seriously.

Whereas Broad draws a sharp distinction between "event" and "becoming", Dewey tries to show how, at this point in history in our understanding of being and time, events ought to be taken to bear traces of variation and indeed temporality "in their inherent structure" [7] (p. 66; cf. [11], Chapter 11, especially pp. 214–216). "Events [for Broad] become but they are not becomings" [7] (p. 65). This makes for an exceedingly odd state of *affairs*. "What is given us by Broad is thus a lot of unchanging [or "eternal"] things, termed, nevertheless, events, with abrupt insertions of changes; time in the usual sense would appear to proceed by jerks or interruptions" [7] (p. 65). That is, time ceases to be a continuum and becomes an unintelligible series of jerky events. In being a continuum, time, on Dewey's view, hardly precludes ruptures or impasses. But this is because *at all levels*, including the most elemental or rudimentary, it is an ongoing continuum which, no matter how finely sliced up, always exhibits qualitative differentiation or variation. As a measurable or tractable affair, temporal order is instituted by means of various instruments, though not necessarily anything precise or even extra-somatic (the observable rhythms of natural processes, including those of one's own breathing or heartbeat, might both be measured and serve as a measure). As a qualitative affair, temporal quality is, however, more primordial than temporal order. A felt sense of qualitative variation is more than a manifest trait of human experience. It is potentially disclosive of nothing less than the traits of nature itself. It thus can serve as a clue for instances of becoming far removed from the narrow sphere of human engagement. To diremept the world into an eternal sphere in which temporal flux (to recall Einstein's expression) is "a stubborn illusion", on the one hand, and a human order in which temporal variation is an ineradicable feature, on the other, commits us to one of the most untenable of dualisms. Unquestionably, this dualism itself proves to be tenacious (its historical hold on the theoretical imagination cannot be denied). But what is manifest in the sphere of our engagements might be true far beyond this sphere— qualitative individuality and temporal quality are not at all illusory.[13] Human experience is no veil or screen falling between human consciousness and the natural world: it is itself a natural phenomenon in and through which other phenomena become available,[14] without inherent limit,[15] to the conscious experience of the querying organism.[16]

Contra Broad, then, there is no distinction to be drawn between event and becoming. Events are themselves eventful. They are also eventualities. The emergent and the eventual, irreducible to antecedent conditions, assume in Dewey's radical temporalism a far greater importance, theoretical and practical, than virtually any theorist in the Western tradition has advocated [3,6,26].

"Events and the Future" is as much a defense of the future as it is a disagreement about the meaning of *event*. On Broad's account, the "future is simply nothing at all" (quoted by Dewey), [7] (p. 65) [27] (p. 70). "A present event is", Broad claims, "defined as one which is succeeded by nothing." Moreover, there is from his perspective no such thing as ceasing to

exist; "what has become exists henceforth for ever" [7] (p. 65; Broad, 68 and 69). For Dewey, however, coming-to-be entails an immanent futurity and, in certain respects, ceasing-to-be involves an irrevocable finality. For instance, the afterlife of Lincoln hardly eradicates the irrevocable fate of his actual death.

The primordial quality of intrinsic variation seems to mean that the structure of *from*, *to*, and *through* defines that of any event. Phasic differentiation or heterogeneity is intrinsic to even the thinnest slice of the temporal continuum. At the most elemental level, we do not encounter eternal, immutable objects or need to posit such objects to make sense out of occurrences taking place at the macroscopic level. Rather, we encounter, at this level as we do at all other ones, qualitatively unique individuals exhibiting a temporal character.

"Broad carries over into his nominal use of the term 'event'", Dewey alleges, "considerations pertinent to modes of thought which attach to what we may call the 'pre-event' era of scientific thought" [7] (p. 66). A distinctively scientific and hence modern understanding of an event is what Dewey is trying to extricate from premodern meanings and implications. If we are successful in this, the bearing of our efforts "upon 'time' and the future should be fairly evident" [7] (p. 66). No distinction is to be drawn between events and instances of becoming. This is indeed a distinction without a difference. So, the bearing of this efforts is, first, to appreciate that "existences are histories or events in the sense of becomings" and, second, if this is so, "past-present-future are on the same level, because all are phases of any event or becoming" [7] (p. 66). Third, "any becoming is from, to, through. Its fromness, or out-of-ness, is *its* pastness; its towardness or intrinsic direction, is *its* futurity; that through which the becoming passes is its presentness". Ontologically, time is no stepchild (or worse, bastard child). Temporally, the future is also not a nullity. "No becoming can be perceived or thought of except as out of something into something, and this involves a series of transitions, which, taken distributively, belong both to the 'out-of' and the 'into', or form a 'through'" [7] (p. 66). What might some readers as a surprising implication of Dewey's understanding of becoming is that, the present "has nothing privileged about it". It is certainly not knife-edged now, exclusive for an immanent from-ness or no less immanent toward-ness (cf. [23] (pp. 573−590, especially p. 574). This can be readily seen by appreciating that "it is as legitimate to speak of the present century or the present geological age as of the present 'moment'". As a phase in a process of becoming, "moment" is an extremely elastic term. It can be stretched to include vast stretches of time, not only the exceedingly narrow phase of the fleeting now. Yet, another surprising implication of these considerations is that "it is a mere fiction that we know pastness and futurity only by means of inference from presentness" [7] (p. 66). Though he does no elaborate his reasons for drawing this implication, Dewey appears to be suggesting that aspects of the present as present might be inferred from those of a known past or an anticipated future (especially a reasonably anticipated future). In any event, he insists, "the 'was', the 'is', when temporally limited to a phase of the going-on, and the 'will be' all stand on the same level *with respect to judgment*" [7] (p. 67; emphasis added). This is evident from our fallibility regarding each phase of becoming: "All of them as judgments are equally susceptible of error". This is because our judgments regarding any of these phases of an instance of becoming "involve inference" [7] (p. 68). And inferences, at least when it pertains to such matters as such phases, always runs the risk of being illicit or unwarranted. "For to say any event, or going-on, has a phase of pastness, presentness, and futurity is not to say *what* has been, is, and will be, is immediately self-revealing" [7] (p. 68). *What* has taken place, *what* is going on, and *what* will *eventually* issue from a process or series of events can only be experientially and, indeed, inferentially determined; and, then, only fallibly.

For a richer sense of what is at stake in "Time and Individual", Dewey's essay must be read in conjunction with *Experience and Nature*, above all, Chapter Three ("Nature, Histories, and Ends"), the one in which he draws the distinction between temporal order (or seriality) and temporal quality. In turn, Dewey's suggestions in *Experience and Nature* regarding this distinction need to be elaborated in the terms articulated in "Events and the Future". An adequate understanding of the focal consideration of "Time and Individuality"—qualitative

individuality—is obtained only by consulting *Experience and Nature*; and, in turn, a firm grasp of his rather elusive conception of "temporal quality" is only obtainable by attending to the details of his critique, in "Events and the Future", of C.D. Broad's conceptualization of events. More needs to be done on both fronts, but for the purpose of our exploration, my hope is that the contours of Dewey's position have been traced in sufficient detail.

### 5. Reprise: Individuality, Ineffability, Intelligibility, and Temporality

Ours is a world of qualitatively distinct individuals, because it is one of irreducibly temporal beings (beings whose very being is becoming). Disregarding these individuals in their qualitative immediacy and using them as prompts for disciplined inquiry, relating *this* to *that* in ways not necessarily given in, or encouraged by, direct experience, marks a decisive advance in experimental investigation. Discarding altogether the qualitative world is, however, neither possible nor desirable. "The world in which we live, that in which we strive, succeed, and are defeated is," Dewey insists in "Qualitative Thought" (1930), "pre-eminently a qualitative world". He immediately adds: "What we act for, suffer, and enjoy are things in their qualitative determinations" [22] (p. 243). Things in their qualitative determinations, however, provide opportunities for conceiving objects in a quantifiable form and, beyond quantifiability, simply in their abstractly relational character. Given the power attained by conceiving affairs as such objects, the reign of mathematical reason has authorized the banishment of qualitative individuality and even of time itself, especially such a thing as "temporal quality" in its inescapably elusive sense.[17] A physicist no less than Einstein was given to pronouncing time "a stubborn *illusion*", a philosopher no less than Bertrand Russell was disposed to insist, "time is an unimportant and superficial characteristic of reality" [28] (p. 215). Cf. [29] (p. 21). In making such pronouncements, Einstein and Russell unmistakably show themselves to be the progeny of Parmenides.

Fixating on temporal order, sequence, or seriality and thereby overlooking temporal quality, immediacy, and individuality does not so much acknowledge the mystery of time as it deprives our sense of temporality of its deep roots in human experience. The difference between humanly scaled time and cosmological time is, after all, a humanly drawn distinction. A just appreciation of the radically diverse scales of conceivable time, however, ought not to license the wholesale discrediting of temporal quality in its most intimate sense.

"The mystery of time is", as Dewey notes, "the mystery of the existence of real individuals" [1] (pp. 111–112). Being is gratuitously given. A necessitarian universe is (in James's words) "a block universe", and Dewey, along with all the other pragmatists, is an uncompromising opponent of the block universe. Being is not externally necessitated, and thus is not logically deducible. It is, on this account, infinitely more than anything else, its own reason for being. It cannot but be begged. It is given so variously and copiously that its status as a gift tends to be obscured, especially to those who desire to *derive* being from something other than itself (e.g., nothing or God).

In "Time and Individuality", then, Dewey not only makes a very strong case for the connection between these terms, but also deftly folds into his discussion a number of other conceptions (seriality, novelty, unpredictability, potentiality, and mystery or, less contentiously, inexhaustible irreducibility). Individuals are identifiable and, to some extent, explicable in a temporal manner, while temporality itself is much more than an external and accidental framework in and through which individuals just happen to move.

The tension between intelligibility and individuality cannot be eliminated. Insisting upon the opacity of individuals in their qualitative immediacy does not make one an obscurantist. However, insisting on the intelligibility of *occurrences* in their indefinite generalizability does not make one a reductivist.

The aesthetic sensibility and the experimental temper need not be exclusive, especially if we appreciate that science is itself, at bottom, an art, specifically, the art of inquiry [1] (p. 268)[18], and in turn, if we appreciate that the arts are inherently experimental (being never anything more than experiments in articulation in which the distinctive quali-

ties of a given medium are explored and indeed exploited for their unique contribution to a given instance of genuinely aesthetic articulation).

"Individuality is", as we have already seen, "at first spontaneous and unshaped; it is a potentiality, a capacity for development". It "develops into shape and form only through interaction with actual conditions" [30] (p. 121). The unique way in which the individual responds to these conditions is always to some extent inexplicable. The qualitatively unique individual and the relationally intelligible phenomenon of that existent ought neither to be conflated nor separated. More must be done to show in detail how this distinction avoids becoming a dualism.

The qualitatively unique constitution of individual existents points to their intrinsic temporality, while the indefinitely intelligible character of them *as occurrences* is indicative of "the [purely] relational character of scientific objects" [1] (p. 105) (i.e., their character as objects of inquiry). Nothing is, however, exhausted in its being by becoming an object of inquiry, just as no individuality is ever fully realized during the bounded career of its temporal duration (paradoxically, its ceasing-to-be becomes integral to its ceaseless coming-to-be, is afterlife a defining feature of its life).

An individual is an individual partly by its inherent resistance to become absorbed without remainder into the environment or setting integral to its emergence and operation. The functional unity of any natural existent is a precarious and transient affair. However, individuality is no less a function of its relationships to others, other than its opaque self-insistence than it is a function of this form of *conatus*. Individuality is more than a nexus of relationships. In any instance, the individual relatum in its qualitative immediacy just is what it is. Its being must be begged. That is, its qualitative uniqueness cannot be logically derived from anything antecedent, especially by an invariant law from an antecedent condition of a bounded or finite constitution.

In the *ongoing* course of at least human lives, at least if we take into account their afterlives (say, the deeds and words of Lincoln today),[19] retrospective resignification is an inevitable feature of what in most other respects are finite lives (e.g., individual lives narrowly bounded by a specific timeframe).[20] The irrevocable character of historical events does not preclude their indefinite resignification. While these events cannot be undone, they cannot but be resignified. However, such resignification is not only retrospective. It is at least as much prospective.

"'This', whatever *this* may be, always implies", Dewey insists, "a system of meanings focused at a point of stress, uncertainty, and need of regulation. It sums up a history, and at the same time opens a new page." *This* makes any of this "record and promise in one; fulfillment and an opportunity". It is "a fruition of what has happened and a transitive *agency* of what is to happen" [13] (p. 264; emphasis added) or what might happen. In countless instances, the system of meanings in place is inadequate for simply identifying the individual affair denoted by *this*. Our attention is called to *this* and, if we are honest about the depth of our uncertainty and the degree of indetermination, we cannot help becoming aware of the inadequacy of our system of meanings, the net of signs presently at our command. New concepts must be crafted ([31]; also, see [32]), in large measured by inherited signs being resignified and, with the aid of such resignification, novel events and phenomena might be more adequately identified and described.

This is precisely what we see in Dewey's own efforts in "Time and Individuality". He is bringing a historical consciousness to a historically emerging sense of time "as a problem". The *problematic* character of temporal flux can only be understood or appreciated in explicitly historical terms. However, his historical narrative directly serves a philosophical purpose and, indeed, a substantive task—that of putting forth an audacious hypothesis regarding the intrinsic connection between time and individuality. In sketching in mostly broad, bold strokes the implications of his hypothesis, Dewey offers a richly suggestive and, in my judgment, inherently plausible account of the mystery of time. Naturalism need not banish a sense of mystery. In fact, the naturalist cannot do justice to individuality without cultivating a sense of this mystery. Our direct encounters with individual existents in their

qualitative immediacy cannot be gainsaid. They are, to some extent, orienting disclosures of what must always remain for us ineffable individuality. Lacking adequate names does not mean lacking adequate means for contextual identification and re-identification of any ineffable individual, contextual identification and re-identification being historical acts in an ongoing engagement in which retrospective and prospective resignifications are pivotal episodes in an interminable drama [11,32].

## 6. Conclusions

If Dewey's remarkable essay truly points us in a promising direction, then any historicist ontology marks a decisive break with the dominant traditions in Western thought. For him as for me, questions of ontology and cosmology are inseparable from methodology. However, methodology needs to be broadly—and, indeed, generously—re-imagined. Giving the arts a possibly equal voice in formulating time as a problem is not only an intriguing, but also a promising suggestion [33]. For the arts no less than the sciences are media of disclosure, exhibiting in their own singular manner traits of nature, thus, aspects of temporality.[21] We need all the help we can get and the power of the arts to illuminate facets of reality that ought not to be dismissed out of hand. This is especially true when we are trying to come to terms with qualitative individuality in its irreducible singularity [1] (pp. 111–114). Our experience of individuals is that of qualitatively unique beings. As such, they are either inherently unintelligible or intelligible in a manner which is fundamentally different from simply subsuming particulars under the grid of categories [34,35].

Our experience of individuals must be granted no less weight than our capacity to explain the appearance, career, and dissolution of individuals. The *haecceity* of individuals indicates a respect in which they are not explicable (perhaps not even cognizable), but one in which they utterly elude identification. Our affectively charged encounters are always, to some extent, disclosive of individuals, just as aesthetically wrought articulations can enhance such disclosures.

The roads to be followed, the discourses to be canvassed, include all of the sites of resignification in which the mystery of ineffable individuality is humbly acknowledged, and the explicability of relational objects is boldly affirmed and ingeniously realized. Relationality, contingency, predictability, and potentiality are in such an approach honored, but so too are immediacy, ineffability, novelty, and *haecceity*. We need a more systematically elaborated account of time as a problem and a mystery, taking into account what has taken place since 1940, the year in which "Time and Individuality" was published. My hope is to have made the case for this exigency, by exploring Dewey's insightful and suggestive essay. To repeat, this essay is itself a site of resignification. Its significance is even more prospective than retrospective, since it points to more to work yet to be taken up than to anything already accomplished. Some of this work has been taken up in earnest by a growing number of daring theorists, including ones in this issue. The vestiges of a commitment to the immutable, invariant, and even eternal are discoverable, often in disguised form. An embrace of the transient, transitional, and simply temporal remains, in many quarters, tentative and partial. A robust affirmation of temporal flux, however, has from at least 1940 to the present been gaining wider acceptance (cf. [11,34,35]). This has been largely the result of efforts devoted to working out the details. While the work of theorists relying mostly on broad, bold strokes, as Dewey does in "Time and Individuality", need to be complemented by such finely detailed work, his approach can never be entirely supplanted. As Whitehead pointed out, the "task of philosophy is to recover the totality obscured by the selection" of details and traits unavoidably emphasized by any theorist. An inevitably vague sense of the whole can serve as a check on an intricately elaborated and detailed account of the cosmos. For the longest time, we have in effect left time itself out of account. In recent decades, our preoccupations with time as a problem have been quite probing and intricate. Even so, much more remains to be done. The process of retrospective—and prospective—resignification is ongoing.

Time as a problem calls for ceaseless resignification and, as such, sets the stage for other occasions for resignification. We need not resign ourselves to such resignifications, for we can embrace it joyfully and unabashedly embrace it as the open-ended adventure it manifestly is. This has increasingly made us appreciate the fateful character of human striving, wherein problems are to be addressed temporally or, more precisely, historically. Historicity must be a part of any answer to our questions, even those arising in disciplines such as mathematics and logic. As C.S. Peirce realized very early in his life, the history of logic is hardly irrelevant to the advancement of that discipline. Formal ingenuity is of the utmost importance; but an informed sense of the actual course of logical thought can aid, in various ways, the cultivation and focusing of such ingenuity. The authority of mathematical or formal reason has been usurped and a commitment to historical and situated reason, of diverse forms, has taken root in widely separated regions [36] (p. 118) [37] (p. 1). Formalism unquestionably has a place—indeed, diverse and multiple loci—but the historical contexts in which purely formal symbolizations acquire their meaning and possess their power cannot themselves be formalized, in any definitive and incontestable manner. Historical contexts can only be understood in historical terms.

"Two hundred years of historical effect have had", Stephen Toulmin observed over fifty years ago, "their effect. Whether we turn to social or intellectual history . . . the verdict is not Parmenidean but Heraclitean" [26] (pp. 355–356; cf. [9,35]). This means acknowledging "nothing in the empirical world possesses the permanent unchanging identity which all Greek natural philosophers (the Epicureans apart) presupposed in the ultimate elements of Nature" [26] (p. 356). In our world, the question can no longer be, "'How do *permanent* entities preserve their identity through all their *apparent* changes'"? It must now be, "'How do *historical* entities maintain their coherence and continuity, despite all the *real* changes they undergo'"? Decades before Toulmin argued for replacing one question with the other, Dewey (and indeed Peirce and James before him) worked strenuously to do just this.

In "Time and Individuality" and other writings, John Dewey addressed this question with a philosophical and historical self-consciousness remarkable for his time [38]. This is one of the reasons why this essay is itself so remarkable. However, above all, it is noteworthy for suggesting *how* this question might be effectively addressed. Although he makes more explicit here, regarding temporality, the presuppositions of his numerous attempts to address specific questions than he does in most of his other writings, the operative presence of these critical presuppositions is evident in all of his major works. Time as a problem points to time and history as solutions. A nuanced appreciation of historicity does not entail a lapse into historical relativism of a self-defeating character. Rather, such an appreciation shows or, at least, suggests how reliance on history itself can provide resources for the transcendence of history, in the only form such transcendence is available to historical actors implicated in such ongoing practices as experimental inquiry and democratic governance, formal education, and psychiatric care. The capacity of a symbol-using and -making animal such as human beings to transcend time is a *temporal* achievement. Their capacity to transcend human history or any other historical process in which they are entangled is moreover a *historical* accomplishment. These capacities, however, never amount to anything more than the attainment of a provisional and precarious perspective, however much they might—possibly must—in some sense count as true, for us at this time [31]. The verdict of history is unequivocal on this. The historical implications of this unequivocal verdict are, however, to some extent, likely to an incalculable degree, unpredictable. John Dewey's approach to the mystery of time suggests no less than this.[22]

**Funding:** This research received no external funding.

**Conflicts of Interest:** The author declares no conflict of interest.

## Notes

1   See, e.g., [2–6].

2   In a very late typescript, composed after "Time and Individuality" [2] (1949–1951), Dewey urges that philosophers adopt what he calls "the empirical, or denotative, method". This method "points out when and where and how things of a designated description have been arrived at. It places before others a map that has been travelled; they may accordingly . . . re-travel the road to inspect the landscape for themselves". This will not make even the most empirically oriented forms of philosophizing into a science, strictly speaking. "The adoption of empirical, or denotative, method would [however] . . . procure for philosophical reflection something of that cooperative tendency toward consensus which markes inquiry in the natural sciences" [13] (p. 389). Even the most self-consciously conjectural and experimental approaches to philosophy will, by virtue of (at least) their generality and reflexivity, operate apart from incontestably scientific forms of experimental inquiry, but they might draw upon, and contribute to, these sciences. Dewey's "Time and Individuality" is a case in point.

3   One of the indications of how prescient Dewey was is that Henri Bergson has been recognized by contemporary theorists as a central figure in our ongoing efforts to fathom the mysteries of time. See, e.g., "Process and Personality in Bergson's Thought" and "What Is Living and What is Dead in the Bergsonian Critique of Relativity" by Čapek [11] (Chapters 5 and 16).

4   Bertrand Helm uses this expression [6] (p. 110). I, however, did not have his use of it consciously in mind when I adopted it to identity Dewey's position.

5   This is language which Dewey uses in "Events and the Future" [7] (pp. 62–68), not "Time and Individuality" [1]. Nevertheless, as I have already suggested, the import of Dewey's assertions in the latter essay can only be adequately grasped in reference to other writings, including "Events and the Future" [7].

6   Here, I am anticipating one of my central claims regarding "retrospective resignification" [18] (p. 79). While this expression is used by Drassinower and those upon whom he is drawing (Barranger et al) in a psychoanalytic context, the term is not limited to the context of its origin. I am confident that this expression has applicability far beyond this context.

7   "Philosophy does its usefulness", A.N. Whitehead asserts in *Process and Reality*, "when it indulges in brilliant feats of explaining away. It is then trespassing with the wrong equipment upon the field of particular sciences. Its ultimate appeal is to the general consciousness of what in practice we experience" [20] (p. 17).

8   Aristotelian physics, with its preoccupation with qualitative alteration, needed to be supplanted by Galilean physics, with its focus on the quantifiable features of natural processes. However, at a certain point, the qualitative world needed to be rescued from its virtually complete erasure by the tyrannical claims of mathematical reason. A number of thinkers in the nineteenth and twentieth century (James, Bergson, Dewey, Heidegger, Whitehead, and of course others) took up the challenge of showing the inherent limits of mathematical reason and, more constructively, our inescapable reliance on qualitative immediacy, not least of all a felt sense of what Dewey would call temporal quality. Dewey's empirical signals a return to Aristotelian "physics" by way of contemporary physics (the physics of his time) in which the quantitative cosmos of theoretical physics is not allowed to eclipse the qualitative world of everyday experience.

9   "The assumption of 'intellectualism' goes contrary", Dewey insists, "to the facts of what is primarily experienced. For things are objects to be treated, used, acted upon and with, enjoyed and endured, even more than things to be known. They are things *had* before they are things cognized" [13] (p. 28; also, see [25], i.e., "Brief Studies in Realism", MW 6, 103–122, especially 111–122). This distinction runs parallel to Martin Heidegger's distinction between things being *zuhanden* (ready-to-hand) and *vorhanden* (present-at-hand) The Deweyan conception of experience being first and foremost *had* is deeply akin to the Heideggerian notion of *Zuhandenheit*. Both philosophers wanted to break the hold of a certain intellectualism without lapsing into obscurantism or, worse, espousing anti-intellectualism.

10  In direct opposition to the dominant tradition in Western thought, Dewey asserts, "the immediately given is always the dubious; it is always a matter for subsequent events to determine, or assign character to. It is a cry for something not given, a request addressed to fortune" [13] (p. 262). Accordingly, any *this* in its immediacy marks simultaneously a felt presence and an elusive identity.

11  As Dewey notes, Broad claims to be following Whitehead, but Dewey is not convinced he does so faithfully (see especially [7] (p. 64, note #3).

12  "If the duration one complete rotation [of an atom] be sliced up into adjacent, successive parts, *the contents of the parts*", Broad asserts in *Scientific Thought* (1923), "*will differ in quality from the contents of the whole*" ([7] (p. 62;) emphasis added by Dewey). The fact that this is alleged for events taking place at the atomic level is of fundamental importance. While Broad goes on to argue for the spatio-temporal *homogeneity* of such contents, Dewey insists upon the qualitative heterogeneity intrinsic to anything properly identified as an "event". The meaning of "event" is indeed at the center of the disagreement here between Dewey and Broad. The opening sentence of "Events and the Future" makes this clear [7] (p. 62). Admittedly, the point of disagreement is a subtle and perhaps elusive one, but we must grasp it; otherwise, we can attain only an inadequate understanding of what Dewey means by "temporal quality".

13　　In *The Timescape of Human Activity*, Theodore R. Schatzki offers a richly detailed and highly suggestive account of the distinctive modes of human temporality, but does not seem to diremp the sphere of human performances from the encompassing domain of natural events. Even so, it might have made the relationship between the two clearer.

14　　As Dewey insists in his *magnum opus,* experience "reaches down into nature; it has depth. It also has breadth and to an indefinitely elastic extent. It strerches" across as well as down [13] (p. 13). He adds, "experience is such an occurrence that it penetrates into nature and expands without limit through it" (ibid.). It is "*of* as well as *in* nature" [13] (p. 12).

15　　"The invention or discovery is", Dewey asserts in *The Quest for Certainty*, "doubtless by far the single greatest event in the history of man. Without them, no intellectual advance is possible; with them, there is *no limit* set to intellectal development except inherent stupidity" [14] (p. 121; emphasis added). In practice, there are countless obstacles and limitations; in principle, there are none—or so Dewey emphatically claims here and elsewhere.

16　　Here, I am using Justus Buchler's term (*query*) to characterize one of the definitive tendencies of the human organism [8] (114f., 142–144). We are irrepressibly given to *probing* in myriad ways, for diverse purposes, whatever we encounter in our experience. Probing is itself a historical temporal process, but no less also a *reflexive* process, since we probe the modes of temporality both *embodied* in processes of probing and *revealed* by them.

17　　In one sense, nothing could be more immediate and undeniable than a felt sense of temporal quality.

18　　Science "is an art" and "art is practice" *and* "the only distinction worth drawing is that not between theory and practice, but between those modes of practice that are not intelligent, not inherently and immediately enjoyable, and those which are full of enjoyed meanings" [13] (pp. 268–269).

19　　On Dewey's account, afterlives are parts of any life, especially the life of such a prominent figure as Abraham Lincoln.

20　　I have borrowed this expression from Abraham Drassinower's *Freud's Theory of Culture*. After quoting "The Infantile Psychic Trauma from Us to Freud" by Madeleine Barranger et al. ("If we were to abide by linear [or unidirectional] categories of causality and temporality, we would be deprived of all therapeutic efficacy"), Drassinower asserts: "Posed on retroactivity, the act of judgment Freud wishes to facilitate is movement of *retrospective resignification*. The erotic struggle against the repetitive intrusions [or compulsions] of the immortal takes place in and through a dynamic act of recollection aimed at historicization" (emphasis added). The ahistoric or "immortal" is displaced by a judgment facilitating historicization. This opens the possibility for reconfiguring the dimensions of time. "For Freud, the future is," Drassinower claims, "not ahead of us but rather in a transvalued relation to the past" [18] (p. 79; see also pp. 151, 162). As I am a pragmatist, I would rather say that the future *can be* ahead of us, in a more open and fulfilling sense than otherwise, only if our relation to the past is transvalued, i.e., only if retrospective resignification proves to be an effective means of altering the basic meaning of irrevocable past.

21　　"If experience actually presents esthetic and moral traits, then these traits may also", Dewey insists, "be supposed to reach down into nature, and to testify to something that belongs to nature as truly as does the mechanical structure attributed to it in physical science" [13] (p. 13).

22　　I benefited immensely from comments, suggestions, and criticisms on an earlier draft offered by Larry Hickman and Jim Garrison as well as the reviewers of this submission, especially one in particular. Finally, Heather Liang's patience and understanding was nothing short of extraordinary, while I struggled for weeks with eye trouble.

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
