# Peer review of "John Dewey’s Radical Temporalism"

_philosophies, doi:10.3390/philosophies8030045_

Round 1

Reviewer 1 Report

The essay is an important contribution to Dewey scholarship. I can't stress this enough.

There are a few type-o's (e.g. line 355 has 2 LW 66 for LW 2, 66) and in one case the missing half of a sentence (fn 10).

Author Response

There is nothing here to which I am asked to reply. The reviewer has finely characterized the purpose of my essay.

Author Response

There is nothing here to which I am asked to respond. Reviewer offers an excellent summation of my enedavor

Reviewer 3 Report

This is a profoundly insightful and well articulated essay on a topic that dives deeply into one of the most complex technical features of Dewey's philosophical project: his radical temporalism. The essay suggests fruitful connections between Dewey's work and the work of other philosophers, including A. N. Whitehead and M. Heidegger. The suggestion that Dewey is more of a full-fledged process philosopher than even Whitehead (whose "'eternal objects' renders time intelligible"), for example, is both remarkable and fecund.  Dewey's idea of "individual as temporal process" is appropriately evaluated as anticipating the work of A. MacIntyre and P. Ricoeur.  Remarkably, both A. Einstein and B. Russell are revealed as "progeny of Parmenides" rather, one gathers, than heirs of Heraclitus.

Readers of the essay might be grateful if the relevant work of each of the authors mentioned in the text (Buchler, e.g.) were included in the list of references at the end of the essay. This would be especially important for C. D. Broad, whose work is extensively discussed but not included, although his book Scientific Thought is listed in the text as quoted by Dewey.

Author Response

There is nothing here to which I am asked to respond except for correction of typos. This has been done

Reviewer 4 Report

Dear authors! Thanks for your interesting discussion.

Let me make a few comments that would help to finalize the article.

1. Introduction. The problem (purpose) of the article is not clear

2. Literary review. Are there any contemporary works that reflect this issue? it is most logical to approach the purpose of the study after analyzing the literature, where you show what has been done and where there are unexplored lagoons. The purpose of the study in this case stems from these discovered lagoons in science.

3. Methods. The methods used by the authors are not expressed.

4. Conclusions. Since the goals and objectives of the article are not set, the conclusions are rather vague.

5. In addition, the article should show directions for future research.

Author Response

This reviewer seems not to have understood the nature of the project. The goals of the paper are very clear: to show how  a relatively neglected essay by John Dewey is relevant to contemporary discussions of time. I can make countless references to more recent literature but this seems beside the point. There is little helpful in this review. The very best Dewey scholars in the world in fact judge this essay to make a major contribution to both our understanding of this figure and his contribution to "the mystery of time." This reviewer is not at all sympathetic with my approach or style. So be it. I feel no obligation to make my paper into what it is not or what it never pretended to be. I respect irreducible philosophical differences. What does not move me is when they are cast as qualitative judgments. 

Reviewer 5 Report

The essay does need another proofread to clean up well over a dozen grammatical errors. There are also two headings listed as 'Conclusion'.  

Author Response

Errors and infelicities have been corrected

Round 2

Reviewer 4 Report

I agree with the author of the article that "to make countless references to more modern literature, but this seems beside the point." I myself often struggle with this requirement, as well as with the requirements for the structure of the article)

However, MDPI journals have adopted an established article structure. It is present even for philosophical journals. This requirement is also pointed out  for reviewers, so it was difficult to bypass it.

Verdict. The article has been revised. The manuscript is ready for publication.

Good luck!